# Impact of a pharmacy-led screening and intervention in people at risk of or living with chronic kidney disease in a primary care setting: a cluster randomised trial protocol

Wubshet Tesfaye [1], Ines Krass,[1] Kamal Sud,[2,3] David W. Johnson,[4,5] Connie Van,[1] Vincent L. Versace [6], Rita McMorrow [7], Judith Fethney [8], Judy Mullan,[9] Anh Tran,[10] Breonny Robson,[11] Sanjyot Vagholkar,[12] Lukas Kairaitis,[13,14] Natasa Gisev [15], Mariam Fathima,[1] Vivien Tong,[1] Natali Coric,[7] Ronald L. Castelino [1,16]

For numbered affiliations see end of article.

**Correspondence to**
A/Prof. Ronald L. Castelino;
ronald.castelino@sydney.edu.au

## ABSTRACT

**Introduction** Chronic kidney disease (CKD) is increasingly recognised as a growing global public health problem. Early detection and management can significantly reduce the loss of kidney function. The proposed trial aims to evaluate the impact of a community pharmacy-led intervention combining CKD screening and medication review on CKD detection and quality use of medicines (QUM) for patients with CKD. We hypothesise that the proposed intervention will enhance detection of newly diagnosed CKD cases and reduce potentially inappropriate medications use by people at risk of or living with CKD.

**Methods and analysis** This study is a multicentre, pragmatic, two-level cluster randomised controlled trial which will be conducted across different regions in Australia. Clusters of community pharmacies from geographical groups of co-located postcodes will be randomised. The project will be conducted in 122 community pharmacies distributed across metropolitan and rural areas. The trial consists of two arms: (1) Control Group: a risk assessment using the QKidney CKD risk assessment tool, and (2) Intervention Group: a risk assessment using the QKidney CKD plus Point-of-Care Testing for kidney function markers (serum creatinine and estimated glomerular filtration rate), followed by a QUM service. The primary outcomes of the study are the proportion of patients newly diagnosed with CKD at the end of the study period (12 months); and rates of changes in the number of medications considered problematic in kidney disease (number of medications prescribed at inappropriate doses based on kidney function and/or number of nephrotoxic medications) over the same period. Secondary outcomes include proportion of people on potentially inappropriate medications, types of recommendations provided by the pharmacist (and acceptance rate by general practitioners), proportion of people who were screened, referred, and took up the referral to visit their general practitioners, and economic and other patient-centred outcomes.

## STRENGTHS AND LIMITATIONS OF THIS STUDY

⇒ This is the first cluster randomised trial to investigate the effect of pharmacy-led screening of chronic kidney disease (CKD).
⇒ The sampling strategy used will ensure recruitment of metropolitan, regional and rural pharmacies to provide representative data.
⇒ Enhanced CKD detection through community pharmacies may also enable earlier commencement of treatment to delay its progression.
⇒ Pharmacist screening may not necessarily generate the anticipated level of referrals and referral uptakes by general practitioners.
⇒ The current community pharmacy reform around 60-day dispensing in Australia may impact recruitment of, and participation by, pharmacies.

**Ethics and dissemination** The trial protocol has been approved by the Human Research Ethics Committee at the University of Sydney (2022/044) and the findings of the study will be presented at scientific conferences and published in peer-reviewed journal(s).
**Trial registration number** Australian New Zealand Clinical Trials Registry (ACTRN12622000329763).

## BACKGROUND

Chronic kidney disease (CKD) is a growing public health problem, which affects over 10% of the adult population worldwide.[1] CKD is defined as sustained kidney damage (presence of proteinuria) and/or reduced kidney function (estimated glomerular filtration rate (eGFR) of <60 mL/min/1.73 m$^2$) lasting for at least 3 months.[2] The global prevalence of CKD is estimated at 9.1%, with high-income countries such as the USA, Canada and Australia the most affected.[1] There has been

a substantial increase in the incidence of people with kidney failure, with people requiring kidney replacement therapies such as dialysis and transplantation increased by 43% over the past 3 decades.[3] CKD is more prevalent in people with socioeconomic disadvantage, ethnic minorities in the UK and the USA, as well as First Nations peoples in Australia, New Zealand and Canada.[1 2]

Inappropriate use of medications in people with CKD ranges from 24% to 70%.[4–9] In Australian primary care, approximately 35% of patients are prescribed potentially inappropriate medications based on their kidney function.[10] As many as 32% of adults with CKD receive inappropriate medications on hospital admission.[11] The inappropriate use of medications in people with CKD can increase risk of adverse drug events, including hospitalisations and death, and many of these events are potentially preventable by optimising their use.[12]

Inadequate recognition of CKD in primary care is one factor that can contribute to the increased risks of adverse drug events. An Australian study reported that although all patients fulfilled the criteria for a standard CKD definition, only 20% of patients had a documented CKD diagnosis.[11] Overall, less than a quarter of patients attending general practices with a pathology report consistent with CKD had a term representing kidney disease in their electronic medical records.[13] Evidence-based guidelines recommend early recognition, referral and management as the most effective approaches to slowing the progression of CKD.[14–17] If CKD is detected early and managed appropriately, loss of kidney function can be reduced by as much as 50%.[16]

The first Australian Strategic Kidney Disease Action Plan was developed with the aim of preventing and managing CKD through effective use of research, evidence and data.[2] As part of this plan, collaboration with pharmacists is proposed as a key strategy to improve patient safety.[4] Community pharmacies represent an underused resource with significant potential to contribute to the early identification and management of individuals at risk of CKD. There are over 5600 community pharmacies spread across metropolitan, regional and remote Australia.[18] Each person in Australia visits community pharmacies 18 times a year (compared with 6 times for general practitioner (GP) visits),[19 20] highlighting the opportunity to reach diverse communities, including those in remote regions with limited access to other health services.

Despite mounting evidence to support pharmacists' involvement in disease screening and medication management,[21 22] community pharmacists currently have limited role in CKD care. Hence, we propose a novel and yet practical intervention to use the community pharmacy setting to screen for CKD using Point-of-Care Testing (POCT) coupled with a quality use of medicines (QUM) intervention. The use of POCT by pharmacists has been previously evaluated and demonstrated in diabetes and in patients taking warfarin.[23–25] Small observational studies from the Netherlands, Spain and Canada also explored the effectiveness of using POCT in

pharmacies for early detection of CKD and identification of nephrotoxic medication use,[26–28] while an Australian study evaluated the feasibility of a pharmacist-led CKD screening.[29]

Our pilot work in an Australian context involving 24 community pharmacies and 389 patients found that 52% of screened participants had moderate or high risk of developing CKD, but only 53% of them subsequently had kidney function tests via a pathology provider.[29] The pilot work highlighted issues such as poor referral uptake by GPs, lack of service remuneration and absence of objective measurements for kidney function.[29 30] To address these challenges, the proposed trial comprises an improved model of care that involves a more effective collaboration between pharmacists and GPs, including incentives for both, and implementing a POCT for objective measurements of kidney function.

## Trial objectives

The overarching aim is to evaluate the effectiveness of pharmacy-led screening in enhanced detection of previously unknown CKD cases and QUM intervention in optimising medication use for patients with CKD. Specifically, the aim is to:

▶ Determine the effectiveness of a community pharmacy-led screening using POCT on the detection of previously undiagnosed CKD compared to risk assessment alone.

▶ Evaluate the impact of community pharmacy-led QUM service informed by POCT in improving the appropriateness of medication use in people with CKD.

▶ Investigate the effectiveness of community pharmacy-led CKD screening and QUM service on long-term economic benefits and humanistic outcomes such as health-related quality of life, and consumer well-being and satisfaction.

## Hypotheses

We hypothesise that the use of a POCT with CKD risk assessment (POCT+QKidney Risk Assessment: Intervention Group) in community pharmacies will result in (1) a significantly improved detection of previously unknown CKD cases, and (2) reduced use of potentially inappropriate medications when compared with CKD risk assessment alone (QKidney Risk Assessment: Control Group).

Other hypotheses related to the primary hypothesis are that the Intervention Group will be associated with a statistically significant increase in the proportions of those who are referred to the GP, take up referral with the GP, are newly diagnosed with CKD compared with the Control Group.

Additionally, the incorporation of POCT in conjunction with the QKidney Risk Assessment will lead to enhanced utilisation of kidney beneficial medications (ie, those that slow kidney disease progression), a long-term economic benefit and improved humanistic outcomes.

## METHODS
### Study setting and design overview

This study protocol is a multicentre, pragmatic, two-level cluster randomised controlled trial. The reporting of this protocol has been guided by the checklist outlined in the Standard Protocol Items: Recommendations for Interventional Trials (SPIRIT statement, online supplemental appendix 1).[31]

Community pharmacies (clusters) are the unit of randomisation and screening participants are the unit of analysis. The project will be conducted in community pharmacies across three Australian states of New South Wales (NSW), Victoria (VIC), and Queensland (QLD) and later expanded to the Australian Capital Territory.

Community pharmacies will be selected from geographical groups of co-located postcodes to form clusters within the top 50 kidney disease hotspot areas identified by Kidney Health Australia.[14] Co-located pharmacies within a cluster represent a local community that will all receive the same service model to reduce contamination bias. Pairs of clusters matched by characteristics such as location and socioeconomic status[32] will be randomly allocated to the intervention or control groups. Objective geographical access is described using the Modified Monash Model (MMM). Consistent with the Australian Government Department of Health and Aged Care's definition used in the Rural Health Multidisciplinary Training Program,[33 34] the MMM for this study was broadly stratified into Metropolitan (MM1) and Rural (MM2–7) categories. Given there were no postcodes classified as 'MM6 – Remote Communities' or 'MM7 – Very Remote Communities' identified within the top 50 kidney disease hotspots, the Rural categorisation was constrained to MM2–5. The number of community pharmacies to be targeted is determined based on the respective population sizes of the states, with proportional pharmacy distributions across Metropolitan and Rural settings. Therefore, based on the population of the states and the percentage contribution of each state, a total of 122 community pharmacies will be recruited from the three states with 1:1 control:intervention allocation, representing 50 pharmacies from NSW (38 Metropolitan vs 12 Rural), 40 from VIC (30 vs 10) and 32 from QLD (20 vs 12). Given the proximity of some of the NSW pharmacies, we have expanded our recruitment to include pharmacies in the Australian Capital Territory.

After the identification of the initial round of postcodes, these were stratified, and colour coded by the geospatial expert (VV) based on socioeconomic status and remoteness to ensure a balanced representation. The allocation into the intervention and control arms was randomised centrally at the University of Sydney by the project manager.

### Selection of community pharmacies

Community pharmacies within identified postcodes will be identified by triangulating sources such as 'Yellow Pages', 'Google' and the pharmacy finder in the Pharmacy Guild's website. All identified pharmacies that are approved to dispense Pharmaceutical Benefits Scheme (PBS) medicines (Australia's universal medicine subsidy programme); have a separate or private counselling area; accredited by an approved pharmacy accreditation programme; have FRED/Z dispensing software will be eligible to participate in the study. Of note, pharmacies without FRED/Z systems but have capacity for manual data entry may be considered for inclusion. Prior to participating in the trial, eligible pharmacies will be required to complete an online training and a baseline survey on existing practices on any screening and QUM services in different chronic diseases including CKD. All identified pharmacies will be contacted by the research team to participate in the study and, if consented to participate, provided with relevant training and onboarding information.

### Sample size

Sample size selection and estimation was performed based on the CKD diagnosis. Using the GRT sample size calculator,[35] it is estimated that the intervention will increase diagnosis from the current CKD prevalence of 10% to between 15% and 20%, an increase of 5%–10%. Sample size is estimated based on a 5% increase in diagnosis of CKD. With 80% power, alpha of 0.05 and intraclass correlation coefficient (ICC)=0.05, at least 57 clusters are required per arm. Based on our sampling model, cluster size has been increased to 61 per arm to account for attrition. With an average number of 30 participants per cluster, the total sample size to be targeted is 3660. Based on previous study[11] that showed a reduction of inappropriate medications in CKD from 32.3% to 17.5% (an approximate reduction of 15%), with 90% power, alpha=0.05 and ICC of 0.05 and a cluster size of 30 per cluster, 19 clusters in each arm are required (38 clusters in all) with a sample size of 1140 for the other primary outcome, that is, change in inappropriate medication use. The sample size required for the diagnosis outcome is, therefore, sufficient to cover the sample size required for the medication changes outcome.

### Selection of pharmacists

Pharmacists who work in the selected community pharmacies will be eligible to participate if:
► They are currently registered by the Australian Health Practitioner Regulation Agency.
► Satisfactorily complete a Continuing Professional Development (CPD)-accredited online training course and assessment.
► Agree to follow procedures outlined in the trial protocol.
► Demonstrate competency in the use of POCT testing using the device supplied for the trial (intervention group).

The content for the CPD online training course, which is accredited by the Pharmaceutical Society of Australia, was developed by the research team for online delivery through Medcast. The online training consists of three or

four self-directed eLearning modules depending on the study arm: (1) trial overview; (2) identification and enrolment of participants; (3) pharmacy-led CKD screening using QKidney risk assessment tool and POCT; and (4) QUM service. Pharmacists in both the intervention (will take all four modules) and control (modules 1–3) groups will be required to complete an assessment at the end of online training, which comprises multiple choice questions and requires an 80% pass mark to obtain a certificate of completion. Pharmacists can claim up to 5 CPD points, with the possibility of claiming additional CPD points for the time spent on conducting services related to the trial. The online learning modules will be evaluated to explore pharmacists' experience and overall level of satisfaction, the relevance of the training to their practice and whether the stated learning objectives were achieved.

Pharmacists in the intervention group will receive additional training on the POCT (performed using Nova Statsensor Creatinine Meter or Nova Max Pro Creatinine and eGFR Meter, Nova Biomedical, Waltham, Massachusetts, USA),[36] devices approved for use by the Therapeutic Goods Administration (online supplemental appendix 2). The analyser in these devices uses a finger prick sample of whole blood (1.2 µL) to measure creatinine within 30 s. In-built software then allows calculation of the eGFR or estimated creatinine clearance via the CKD Epidemiology Collaboration or Cockcroft-Gault equations. Such POCT devices are used to determine kidney function used in radiology and cardiac catheterisation services to minimise risk of contrast-induced nephropathy as well as in oncology and emergency departments to expedite patient care. They have also been used in emergency and community pharmacy settings for detection of nephrotoxic medications.[28 37] All participating pharmacies will receive the online training modules specific to the intervention group including on POCT and are expected to demonstrate competence in obtaining the blood sample using the device.

### Selection of participants eligible for screening

Participants who are eligible for screening must:

▶ Be between the age of 35 and 74 years.
▶ Be at an increased risk of developing CKD due to one or more of the following parameters: ≥60 years of age, hypertension, diabetes, established cardiovascular disease, obesity (body mass index (BMI) ≥30 kg/m$^2$), current or former smoker, family history of kidney disease, prior acute kidney injury, Aboriginal & Torres Strait Islander origin.
▶ Not have kidney failure or not be receiving kidney replacement therapy.
▶ Not have a terminal illness.
▶ Not be pregnant.
▶ Be able to make independent decisions about their health.

Eligibility will be determined based on self-report by the prospective screening participant and based on the

pharmacist's judgement of the participant being able to make independent decisions about their health.

## CLINICAL PROTOCOL AND RECRUITMENT PROCEDURE

The clinical protocol for the trial is summarised in figure 1, which illustrates each of the steps involved in recruitment, screening, QUM service and referral to GP. Promotional materials including flyers, posters and eligibility card for self-assessment will be provided to each participating pharmacy. Each pharmacy will assign one person to serve as the study champion, while all pharmacy staff members, including pharmacists, pharmacy interns and pharmacy assistants and technicians will be encouraged to support the recruitment. The expected duration of the participant recruitment is 6 months per pharmacy. The POCT kit in the intervention group pharmacies will be rotated across pharmacies over 12 months to ensure each pharmacy has at least 6 months to recruit 30 participants.

Screening participants will be recruited through the targeted community pharmacies. The recruitment process involves the pharmacists discussing the study and confirming participant eligibility. The pharmacist will manage potential participants using a purpose-built online software application hosted on a cloud-based server at The University of Sydney. This will ensure the integration of the recruitment process, including the delivery of intervention into the pharmacy recording platform and thereby automate key processes, such as risk assessment, medication history taking, referral letter production and follow-up. Also, direct storage of data in the cloud server will ensure no data are lost due to any systemic issues arising from individual pharmacies.

Following identification of screening participants, those who are eligible and willing to participate will be provided with the participant information statement to read and be asked to sign two consent forms. The first one is to participate in the study and to access their medical records from the GP and My Health Record (a secure online platform used in Australia used to store key healthcare information—this will facilitate pharmacists' verification of clinical information when they make referral to GPs). The second consent will be used to release their Medicare Benefits Schedule (MBS) and PBS claims data from Services Australia.

Following recruitment, screening participants in the _**control**_ group, if they are known patients with CKD (stages 1–4; based on self-report), will be provided with lifestyle advice and counselling on medications to avoid in kidney disease along with a Kidney Health Australia factsheet on 'Looking after yourself with kidney disease'. All other control group patients will have their risk of developing CKD computed using the QKidney risk calculator,[38] or based on blood pressure levels, if they have at least one of the known CKD risk factors. Participants will be advised of an appropriate action based on their calculated risk. All participants will be provided with lifestyle advice using

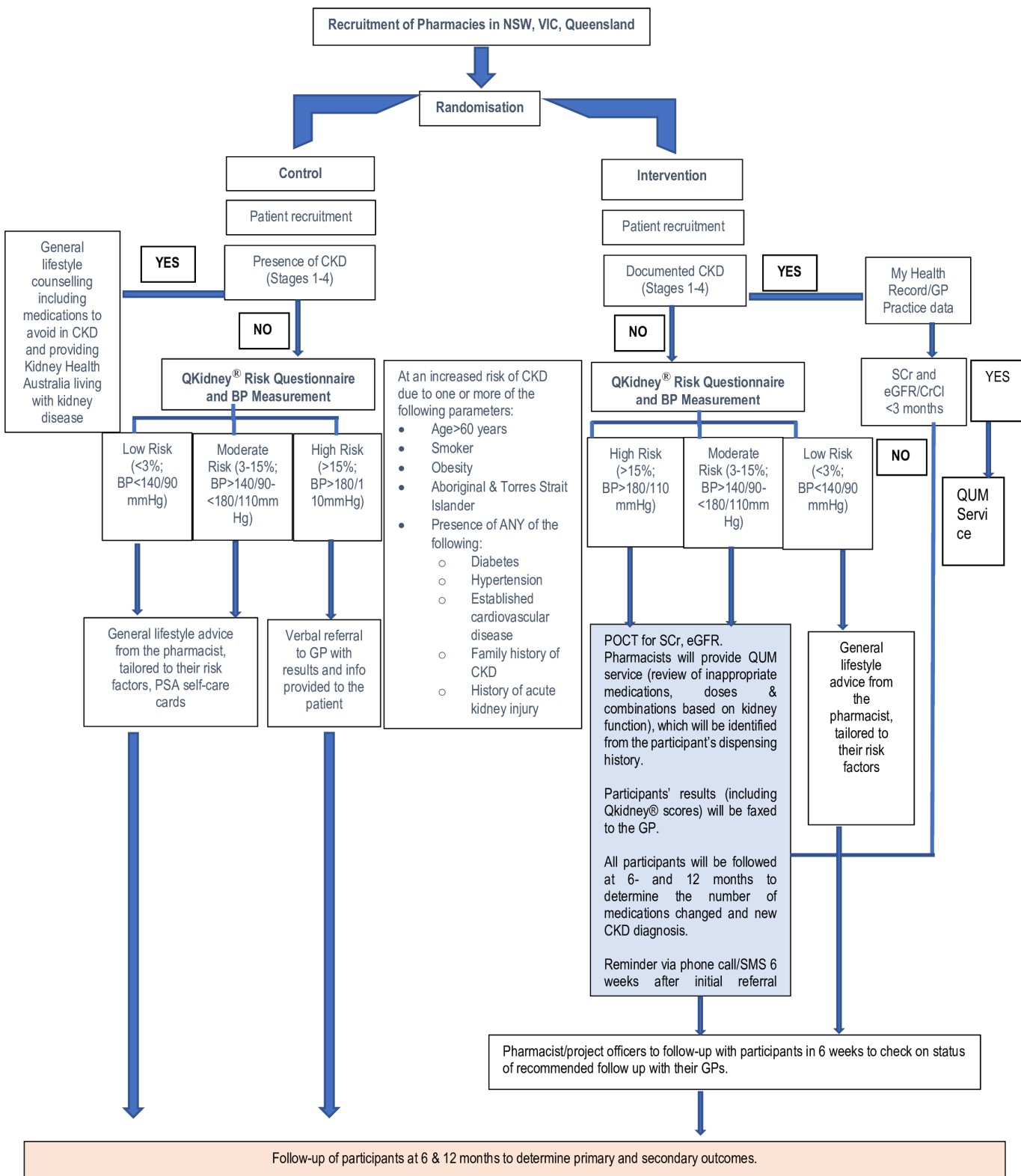

**Figure 1** Procedures of recruitment of participants. BP, blood pressure; CKD, chronic kidney disease; CrCL, creatinine clearance; eGFR; estimated glomerular filtration rate; GP, general practitioner; NSW, New South Wales; POCT, Point-of-Care Testing; PSA, Pharmaceutical Society of Australia; QUM, quality use of medicines; SCr, serum creatinine; SMS, short messaging service; VIC, Victoria.

a range of educational fact sheets, sourced from Kidney Health Australia resources, targeting their risk factors for CKD.

Following recruitment, participants in the *intervention* group will be divided into two groups.

1. Those with pre-existing CKD (Stages 1–4; based on self-report) and a serum creatinine (SCr) and eGFR measured within the last 3 months (accessed via My Health Record) will have their dispensing history and medication event summaries (My Health Record) reviewed to determine the presence of any medications considered problematic in kidney disease. Participants with inappropriate medication, doses or medication combinations based on their kidney function will have recommendations to adjust the doses of medications or cease contraindicated medications (QUM service) to their GP. Additionally, participants who, because of their kidney function, are eligible but not receiving potentially beneficial medications (eg, renin angiotensin system inhibitors or sodium glucose co-transporter 2 inhibitors) will have recommendations provided to their GPs to consider starting these medications. The pharmacists will be encouraged to upload their recommendations on the My Health Record platform as event summaries. Participants with no SCr/eGFR recorded in the last 3 months will have their kidney function determined using POCT followed by the QUM service.

2. Participants with no prior history of CKD and at least one CKD risk factor will have their risk of developing CKD computed using the QKidney risk calculator. Participants who return with a moderate to high risk (>3% of developing moderate to severe CKD over 5 years) will have their kidney function determined using a POCT device for SCr and eGFR and will have their dispensing history reviewed to determine the presence of any medications considered problematic in kidney disease. Participants with any inappropriate dosages or contraindicated medications as per their kidney function will have recommendations to their GP to adjust the doses of medications/ceasing contraindicated medications or medication interactions (QUM service). Also, the recommendations may include introduction of potentially beneficial medications as highlighted above. The pharmacists will be encouraged to upload their recommendations on My Health Record as event summaries.

Screened participants in the intervention group are expected to attend a pharmacy session lasting between 20 and 45 mins with the trained pharmacist to assess and collect basic demographic and clinical/medical information (eg, height, weight, blood pressure, medications dispensed), and to apply the QKidney Risk assessment and POCT protocol and operating procedures. The time it takes for the overall procedure depends on the individual participant's circumstances, such as the presence of pre-existing CKD or level of risk of developing it, type and number of medications involved as well as the need for screening referral to a GP. Similarly, participants in the control group will be expected to attend a pharmacy session at baseline lasting between 20 and 30 mins for the relevant risk assessment using the QKidney Risk tool and to provide relevant education and resources pertinent to their level of risk and risk factors for CKD.

Participants in the intervention group will be followed up by pharmacists/research project officers via phone or SMS (short messaging service) at 6 weeks to determine the status of the recommended follow-up with the GP and to re-advise GP review if participants have not yet made an appointment with their GP. All participants will be followed-up by pharmacists/research officers at 6 months and 12 months to determine the number of medications changed and new cases of CKD diagnosed (in participants with no prior history).

## Data collection, monitoring, management and reporting

Data collection is purpose designed and will be facilitated by a highly secured and password protected online software application. Study protocols, templates for assessing CKD risk, links to the QKidney risk calculator, template for GP referral, QKidney health advice, template for QUM service will be included in the software along with specific guides to support access and familiarisation. Intervention group pharmacies will also be able to record the POCT results in the software. This will maximise trial efficiency and reduce the burden on pharmacists.

Adherence to the study protocol will be regularly monitored for accuracy and completeness by the project manager and research officers using the online study website and site visits. In addition, project officers will use a range of communication strategies to ensure that the community pharmacists are kept informed of the latest developments of the study, and to provide ongoing support to the pharmacists and these include regular phone calls and bulk emailing options to allow an early action where pharmacies require additional support with the study.

Data collection including pre-screening, screening and post-screening activities and the respective time points are outlined in table 1. The study is set to take place over 2 years (March 2023 to May 2025), and will include enrolling pharmacists and patients, 6-month and 12-month follow-ups and report writing.

## Outcomes
### Primary outcome measures
► Proportion of patients newly diagnosed with CKD at the end of the study period.
► Changes in the number of medications considered problematic in kidney disease (number of medications prescribed at inappropriately high dose, number of contraindicated medications or detrimental drug interactions).

### Secondary outcome measures
► Proportion of screening participants with inappropriate medications (number of medications at inappropriate high dose based on the kidney function and number of contraindicated medications).

**Table 1** Study procedures and time points

| Procedure | Baseline | 6 weeks | 3 months | 6 months | 12 months |
|---|---|---|---|---|---|
| Pre-screening/screening activities | | | | | |
| Site randomisation | X | | | | |
| Information sheet | X | | | | |
| Pharmacist/participant consent | X | | | | |
| Initial screening | X | | | | |
| Pharmacy survey (current services) | X | | | | |
| Inclusion/exclusion criteria - review | X | | | | |
| Demography and medical history | X | | | | |
| Outcome assessment | | | | | |
| Proportion of people with CKD | | | | X | X |
| Changes in # of PIMs | | | | X | X |
| % of participants with CKD risk factors | X | | | X | X |
| QKidney risk profiles | X | | | | |
| % Participants referred to GP | X | | | | |
| % Participants taking up referral with GP | | | | X | X |
| Phone/SMS reminder (PO/RA) | | X | | | |
| Other outcomes (health-related quality of life, well-being) | X | | | | X |
| Study evaluations | | | | | |
| Economic evaluation (health resource use and costs) | | | | | X |
| Pharmacist midtrial Interview | | | X | | X |
| Pharmacist end of trial interview | | | | | X |
| GP satisfaction | | | | | X |
| Participant knowledge/satisfaction survey | | | | | X |
| Study compliance audit | X | | X | X | X |

CKD, chronic kidney disease; GP, general practitioner; PIMs, Potentially Inappropriate Medications; PO, Project Officer(s); SMS, short messaging service.

► Proportion of eligible participants without potentially beneficial medications (eg, renin angiotensin system inhibitors or sodium glucose co-transporter 2 inhibitors).

► The number and characteristics of proposed medication adjustments recommended by pharmacists and acceptance rates by GPs.

► Proportion of participants with CKD risk factors.

► Proportion of participants screened (process indicator).

► Participants' QKidney risk profiles.

► Percentage of participants referred to their GP.

► Percentage of participants who take up referral with their GP.

► Number of hospitalisations due to medication-related events.

► Cost effectiveness.

► Health-related quality of life, consumer well-being and satisfaction.

Both the intervention and control group participants will be followed-up by the project officers/pharmacists at 6 and 12 months to determine if they have had any additional blood tests, and changes to their medications, and hospitalisations.

Primary outcomes will be determined based on data collected from GPs. Health-related quality of life and overall participant well-being will be measured using the EuroQoL-5D-5L, which consists of a descriptive system and a Visual Analogue Scale.[39] The EQ-5D-5L is scored on a 0–1 utility scale, where 0=death and 1=full and complete health. Unit values will be informed by Australian population estimates.[40] A minimally important difference of 0.07 (7%) change in utility on the EQ-5D-5L scale is considered to be clinically meaningful.[41]

Screening participants' satisfaction with the service and willingness to pay for the service will be determined by a telephone questionnaire, which will include items relating to service approval (5-point Likert scale); and open-ended questions to elicit reasons for approval and satisfaction ratings. Screening participants' willingness to pay for the service will also be determined including the maximum amount. Satisfaction of pharmacists (at midtrial and end of trial) with the screening and QUM services will be assessed using electronic surveys. The GPs

involved in the care of the participants will also be sent a survey at the end of the trial to determine their satisfaction and any effects on their practice.

A purposive sample of screening participants and GPs will be invited for a follow-up interview based on their responses to the surveys. The aim will be to identify issues that may limit the effectiveness of the proposed intervention. Pharmacist satisfaction with the service will be determined by inviting pharmacists who delivered the intervention to attend focus group discussions. The topics covered will include overall pharmacist experience of the screening programme, including consumer perspectives, GP communications, business impact and implications for future implementation.

### Data analysis plan
All data will be reported following the Consolidated Standards of Reporting Trial guidelines.[42] Baseline clinical and demographic characteristics will be summarised using frequencies and percentages for categorical variables, and means, SD, medians, IQRs and minimum and maximum values for continuous variables, as appropriate. All statistical analyses for the primary and secondary outcomes will follow the intention-to-treat (ITT) principle. The ITT population will include all randomised participants based on their initial assignment to either intervention or control group and include any participants who prematurely withdraw or poorly comply with the protocol.

All primary and secondary analyses will use Generalised Estimating Equations with pharmacy as the unit of clustering, using the model appropriate for the distribution of the outcome. The analysis of the primary outcome (reported as a binary variable 0=not diagnosed with CKD, 1=diagnosed with CKD) will use a logistic regression within a model to test the effectiveness of the intervention. Analysis will first be unadjusted (including as fixed factors group membership and the stratification variable, region), followed by analysis adjusted for relevant baseline covariates (age, gender, blood pressure measures, BMI, socioeconomic status and location of pharmacies), as well as the fixed factors from the unadjusted analysis. Binary secondary outcomes such as percentage of participants with CKD risk factors, those having a GP review, and those with potentially inappropriate medications will be analysed using a similar approach as that for the primary outcome.

Additional outcomes including numbers of recommendations made by pharmacists, numbers of medication-related hospitalisations or acceptance rates by GPs will be analysed using negative binomial regression (or Poisson regression, depending on distribution of the outcome) or logistic regression and will include unadjusted and adjusted analyses as per the primary outcome. For humanistic outcomes such as participant's health-related quality of life, as well as outcomes on GP satisfaction and participant knowledge/satisfaction, we will use linear regression. All analyses will be presented using

unadjusted and adjusted models as per the primary outcome. For the proportion of participants screened in the trial (process indicator), we will use the Mantel-Haenszel extension of a $\chi^2$ Test of Independence.

The proportion of missing data will be determined and assessed for any patterns related to the missingness. Unless otherwise stated, multiple imputation (MI) will be used to impute missing values. Missing dichotomous data and categorical data with >2 categories will be assessed using the appropriate MI regression techniques, and missing continuous data will be imputed using chained equations. Sensitivity analyses to explore the effect of imputation will be conducted (eg, complete case; assuming worst case for the missing outcome observations). All tests will be two-sided, with statistical significance set at 0.05 and we will use IBM SPSS Statistics for all analyses unless otherwise specified. There will be no adjustment of the alpha level to account for multiple comparisons because the primary and secondary outcomes are specified, unless otherwise stated to be post hoc.

### Study evaluation
A logic model for evaluation of this study is presented in figure 2. The model summarises the activities involved in the development and implementation phases, as well as the projected outcomes. The model will be used to frame our approach to evaluating (1) whether the trial was developed and implemented as intended (fidelity and reach), (2) whether study outcomes were achieved or not, (3) the influence of situational factors on results obtained. Baseline surveys, results from the screening and QUM services and pharmacists' interviews will be used to assess these activities.

### Baseline evaluation
At the beginning of the study, participating pharmacies will be required to complete a baseline survey that includes questions on pharmacy characteristics, existing professional services (remunerated and non-remunerated) and strategies and capacity for implementing these services. The information collected will be used to inform the comparability of the two arms in the trial and health economic analysis. All data entered by participant pharmacists into the study software will be extracted to generate quantitative data to enable the trial process evaluation.

### Impact evaluation
The impact of the screening and QUM services will be determined by testing the clinical hypothesis as detailed in the clinical protocol and outcomes sections.

### Midtrial interviews
Three months after the implementation of the service in the pharmacy, a mix of pharmacists from high and low performance groups in both arms will be invited to participate in an interview to provide feedback to identify challenges, opportunities, barriers and facilitators to service

| INPUTS | ACTIVITIES | OUTPUTS | SHORT-TERM OUTCOMES | ANTICIPATED LONG-TERM OUTCOMES |
|---|---|---|---|---|
| **User and Stakeholders** Consumers (patients) Community pharmacists Community pharmacy staff GPs and practice staff<br><br>**Technology Components** Electronic templates for screening and QUM services, including referral templates, Qkidney® risk and results.<br><br>**Training** Online CPD training program developed for participating pharmacists.<br><br>**Other resources** Material resources (e.g., promotional material (flyers, posters for display etc), Stat sensor POCT for SCr and eGFR/CrCL. | **Preliminary** Project steering committee and Expert panel convened to provide advice on trial implementation; and how to support engagement with GPs, Primary Health Networks, recruit, retain participants, and community pharmacies.<br><br>Recruit pharmacies.<br><br>Implementation of the cRCT.<br><br>**Pharmacists' intervention** Initial training. Implementation planning. Recruitment of eligible participants. Referral of participants with undiagnosed CKD and/or inappropriate medications as per kidney function to the GP according to the clinical protocol.<br><br>Follow up of patients at 6 and 12 months. Regular report of trial progress.<br><br>**Maintenance** Maintain relationship with pharmacies, pharmacists, and other stakeholders. | Improved rates of detection of people with undiagnosed CKD<br><br>Improved use of medicines in patients with kidney disease<br><br>Improved awareness of lifestyle measures to reduce CKD risk.<br><br>Improved collaboration between pharmacists and GPs | Will a pharmacy-led screening and QUM service improve identification of undiagnosed CKD and improve the use of medications?<br><br>**Primary Outcomes** Proportion of newly diagnosed CKD patients<br><br>Proportion of medication changes<br><br>**Secondary Outcomes** Proportion of people with inappropriate medications<br><br>Proportion of people with chronic kidney disease<br><br>Proportion of people screened.<br><br>QKidney® risk profiles<br><br>% Of people referred to their GP<br><br>% Who take up referral with their GP.<br><br>Hospitalisations due to medication-related events.<br><br>Quality of life and other humanistic outcomes such as well-being and consumer satisfaction.<br><br>Cost effectiveness. | Reduced incidence of kidney failure.<br><br>Reduced medication-related events, hospital admissions and mortality related to CKD.<br><br>Guide health policy to efficient use of primary health care resources. |

**Figure 2** Pharmacy based screening and QUM study logic model. CKD, chronic kidney disease; CPD, Continuing Professional Development; cRCT, cluster randomised controlled trial; CrCL, creatinine clearance; eGFR; estimated glomerular filtration rate; GP, general practitioner; POCT, Point-of-Care Testing; QUM, quality use of medicines; SCr, serum creatinine.

implementation. This will enable us to provide support to pharmacies and adapt where appropriate.

### End-of-trial interviews
At the end of the study, pharmacists from participating pharmacies representing a mixture of both high recruitment and low recruitment, will be interviewed to gain feedback on the service implementation in the pharmacy, identifying barriers and facilitators to implementing the trial in the pharmacy, perceptions of sustainability and level of engagement by the local GPs in the referral process.

All study participants will receive a survey to determine their quality of life and well-being. Referred participants will also receive a survey to determine their level of satisfaction and willingness to pay for the provided service. A random sample of 20% non-referred participants will be surveyed to determine their overall experience and level of satisfaction with the service. A random sample of 10% of GPs who had patients referred to them during the trial will be invited to respond to a telephone/electronic questionnaire to determine their overall experience and level of satisfaction with the service.

All surveys will be created on Research Electronic Data Capture, a secure web-based application designed to support data capture for research studies hosted at The University of Sydney.[43] Qualitative interviews will comprise open-ended questions that prompt discussion on a wide range of topics including overall experience with the study, consumer feedback, interaction with GPs and impact of the trial in their pharmacy.

### Economic evaluation
Health service utilisation will be measured through Medicare and Pharmaceutical Benefits Schemes (MBS and PBS) data linkage, to quantify the symptom-specific and total health service resource use and costs at 12 months, by allocated group. The Consolidated Health Economic Evaluation Reporting Standards guidelines (online supplemental appendix 3) will be followed when reporting the health economic evaluation.[44]

### Costs
Cost-effectiveness of the intervention will be calculated from a health service provider perspective considering costs incurred by the pharmacists and the government (PBS/MBS) but will not include costs borne by screening participants (eg, through co-payments charged by GPs). Healthcare resource use including GP visits, specialist visits, diagnostic tests, medicine, hospitalisation, emergency admissions will be collected from MBS/PBS history, medication prescription and dispensing records and previous hospital discharge summaries from My Health Record. Unit costs will be collected based on MBS/PBS item numbers, Australian Refined Diagnosis Related Groups and Triage codes. Measured costs will

include pharmacist time to facilitate data collection at each pharmacy.

### Costs analysis

The cost per new CKD case diagnosis for each group will be estimated and compared. We will assess whether the average cost of the new CKD case diagnosis in the intervention group is lower or higher than the average cost of the new CKD case diagnosis in the control group. We hypothesise that there is likely to be a difference in average costs per new CKD case diagnosis across the sites in the same cluster and across the clusters from the metropolitan and rural settings. This is due to the differences in sample size, local unit prices of equipment, staff costs and different established local practices.

Differences in overall mean costs between the two groups will be analysed using Generalised Linear Mixed Model. The distribution of residuals from the regression model will be examined and a decision will be made as to whether the costs should be logarithm transformed. Any extreme costs will be carefully examined to decide if they should be considered as extreme outliers and should be trimmed (removed) from the analysis. It is anticipated there will be some missing data. Missing costs data will be examined to ascertain whether it is missing completely at random, missing at random or missing not at random. Simple and multiple imputation methods will be considered based on the pattern of missingness.[45]

### Effectiveness

Based on the EQ-5D-5L, utility weight will be calculated via an Australian algorithm, which will be used to obtain quality adjusted life-years (QALY).

### Within-trial cost-effectiveness analysis

Incremental cost effectiveness ratios that allow analysis of incremental costs of intervention group versus control group per one QALY gained will be calculated.

### Modelled cost-effectiveness analysis

The modelled economic evaluation would investigate the benefits of early diagnosis of CKD cases and the associated prevention/delay of the CKD complications like use of kidney replacement therapies or mortality in the long term. The modelled economic evaluation will include the long-term costs of treatment, regular laboratory investigations, treatment of adverse events, hospital admissions and other relevant costs.

The long-term improved outcomes are hypothesised to translate into a cost reduction and offset the increased costs associated with adding pharmacy-led screening to no screening, for CKD case diagnosis. The modelled economic evaluation will also use published results of long-term observational studies to derive assumptions for the decision analytical modelling and to populate the model. A 5% discount rate will be applied to both costs and benefits.[46] The uncertainty about results of economic evaluation will be explored using a non-parametric bootstrapping method.

### Patient and public involvement

The study will engage key stakeholders in designing, conducting and reporting the research, as shown in figure 2. Input from consumers and consumer organisations will be actively sought throughout the study implementation. This will be achieved by involving consumers and organisations, such as Kidney Health Australia and the Pharmaceutical Society of Australia, in both the project steering committee and the expert advisory group. Their roles will include contributing to the development of training materials for pharmacists, advising on effective communication and engagement strategies with pharmacists and GPs, and facilitating the implementation and dissemination of research progress and findings via newsletters.

### Ethics and dissemination

The study will be conducted in compliance with the International Conference on Harmonisation Good Clinical Practice guidelines, the World Medical Declaration of Helsinki and all amendments and the Protocol and Protocol Amendments.

The protocol, participant information sheets, consent forms and other relevant case report forms have all been approved by The University of Sydney Human Research Ethics Committee (2022/044). The trial protocol is also registered with the Australian New Zealand Clinical Trials Registry.[47] The trial registration is provided in online supplemental appendix 4. Study findings will be disseminated in plain language and peer-reviewed forums, including at conferences, via publication in medical journals, and to participants, study staff, clinicians and patient groups, via direct approaches and a variety of traditional and electronic media.

## DISCUSSION

The proposed study presents a protocol for the first large-scale cluster randomised trial to evaluate the effectiveness of community pharmacy-led CKD screening and medication management in improving early disease detection and QUM. The sampling plan primarily targets kidney disease hotspot areas identified by Kidney Health Australia, but also accounts for access to services (ie, metropolitan and rural locations). Within Australia, socio-economic status is lower in rural and remote areas relative to metropolitan areas.[33] The inclusion of MM2–MM5 locations is also relevant to First Nations communities—these areas overlap much of Inner and Outer Australia where 44% of First Nations people reside.[33 48] The study is expected to raise awareness among both pharmacists and at-risk individuals about CKD and its risk factors, potentially leading to the early detection of previously unknown CKD cases. Further, by promoting the safe and effective use of medicines, the study findings will potentially lead to reduced adverse drug events and medication-related hospital admissions in people at risk of, or who live with, CKD. Collectively, the study will generate evidence to support the translation

of new preventive and public health measures into practice and implement an innovative approach to improving the quality and cost-effectiveness of preventive healthcare intervention.

Effective and sustainable primary care interventions aimed at the early detection and management of CKD have the potential to not only improve health outcomes but also significantly reduce costs associated with extensive hospital services required in later stages, mainly due to the need for regular dialysis or a kidney transplant, which is estimated to cost the Australian economy $A5.7 billion a year.[49] More importantly, recent modelling has shown that investing in early detection of kidney disease can save the Australian economy about $A10.2 billion over the next 20 years.[49] Pharmacists are well placed to play a role in the early detection, education and referral of individuals at elevated risk of CKD. By offering screening and QUM services, community pharmacists can contribute to health promotion and education. As integral members of primary healthcare, community pharmacists serve as an important channel for delivering these types of services. Further, their involvement in CKD care contributes towards improving the health status of Australians, including those from lower socioeconomic groups who are disproportionately affected by CKD; increasing public knowledge and awareness of CKD and its risk factors; expanding public health roles for community pharmacists; and enhancing the roles of pharmacists as an integral part of the health workforce. This aligns with Australia's Primary Healthcare 10-Year Plan, which promotes better utilisation of primary healthcare workforce, including allied health professionals and pharmacists.[50]

There is widespread agreement that an effective programme is needed to support the identification and better management of individuals with early stages of CKD. The subsequent adoption of such a programme, if shown to be successful on a larger scale and in routine practice in Australia, has enormous potential to reduce the population levels of risk and hence reduce preventable healthcare expenditures due to the need for expensive treatments in advanced disease. The adoption of the programme will also enhance the effectiveness of currently remunerated pharmacists' QUM services such as MedsCheck and Diabetes MedsCheck, which aim to improve patients' understanding of their medications and ultimately patient outcomes.

By targeting a representative sample of community pharmacies matched by location (rurality) and socioeconomic status,[51] the study is anticipated to represent a wide range of demographics and regions in Australia. Some of the limitations anticipated during recruitment have been previously described and, while it may not be possible to avoid them fully, we will implement proper incentives to the participating pharmacies, pharmacists and GPs. One major limitation of implementing such programmes is the lack of kidney function information availability, which impacts pharmacist's ability to identify and intervene in medication-related problems in people with CKD. Using POCT for determining kidney function markers in community pharmacies, the proposed study aims to fill this gap and promote pharmacists' contributions to CKD care. Finally, the high staff turnover rates and uneven distribution of the pharmacy workforce based on remoteness may affect the recruitment of pharmacists and patients in certain settings, particularly in remote and rural areas.[52 53]

This trial introduces an innovative approach by leveraging existing healthcare resources like My Health Record for a more streamlined referral and medication management purposes. This will improve the utilisation of available resources and maximises their potential. Further, the trial employs an integration-based software system that works with selected pharmacy dispensing systems. This integration is expected to provide a more efficient decision support system for medication review and referral, contributing to improved overall patient care. In addition to exploring community pharmacies' readiness to incorporate technological solutions to address QUM issues, the study will shed light on potential barriers and enablers to scalability and sustainability of such systems.

In summary, this research presents a unique opportunity for community pharmacists to engage in health promotion and education of people who are at risk of or living with CKD. Given current existing chronic disease services in community pharmacies, our research will incorporate pharmacists in CKD screening and ensure quality medicines use in people who are at risk of or living with CKD. This is in line with the recommended clinical practice guidelines and National Strategic Plan that highlight the importance of early identification and intervention to slow the progression of CKD and occurrence of complications through a multidisciplinary primary care approach.[2 54]

**Author affiliations**
[1]The University of Sydney School of Pharmacy, Sydney, New South Wales, Australia
[2]Nepean Kidney Research Centre, Department of Renal Medicine, Nepean Hospital, Sydney, New South Wales, Australia
[3]The University of Sydney School of Medicine, Sydney, New South Wales, Australia
[4]Centre for Health Services Research, The University of Queensland Faculty of Medicine, Brisbane, Queensland, Australia
[5]Department of Kidney and Transplant Services, Princess Alexandra Hospital, Brisbane, Queensland, Australia
[6]Deakin Rural Health, School of Medicine, Faculty of Health, Deakin University, Melbourne, Victoria, Australia
[7]Department of General Practice and Primary Care, The University of Melbourne, Melbourne, Victoria, Australia
[8]School of Nursing, The University of Sydney Susan Wakil School of Nursing and Midwifery, Sydney, New South Wales, Australia
[9]Graduate School of Medicine, University of Wollongong, Wollongong, New South Wales, Australia
[10]NHMRC Clinical Trials Centre, The University of Sydney, Sydney, New South Wales, Australia
[11]Kidney Health Australia, Melbourne, Victoria, Australia
[12]MQ Health General Practice, Macquarie University, Sydney, New South Wales, Australia
[13]Department of Renal Medicine, Blacktown Hospital, Sydney, New South Wales, Australia
[14]Western Sydney University School of Medicine, Sydney, New South Wales, Australia

[15]National Drug and Alcohol Research Centre, University of New South Wales Sydney, Sydney, New South Wales, Australia
[16]Pharmacy Department, Blacktown Hospital, Sydney, New South Wales, Australia

**Contributors** WT and RLC drafted the manuscript. WT, IK, KS, DJ, CV, RM, JM, AT, SV, LK, NG and RLC were involved in the study concept and design. VV developed the geographical sampling frame and randomisation postcodes. JF developed the statistical analysis plan, while AT developed the health economics evaluation plan. WT, IK, KS, DJ, CV, VV, RM, JF, JM, AT, BR, SV, LK, NG, MF, VT, NC and RLC critically reviewed the trial protocol. All authors have read, revised and approved the final manuscript.

**Funding** This work was supported by the Medical Research Future Fund of the Australian Government primary healthcare initiative (MRFQI00008) and conducted collaboratively between The Universities of Sydney, Melbourne, Queensland, and Kidney Health Australia. The funding body has no role in the design of the study as well as collection, analysis and interpretation of data and manuscript write-up.

**Competing interests** DJ has received consultancy fees, research grants, speaker's honoraria and travel sponsorships from Baxter Healthcare and Fresenius Medical Care, consultancy fees from AstraZeneca, Bayer and AWAK, speaker's honoraria from ONO and Boehringer Ingelheim and Lilly, and travel sponsorships from Ono and Amgen. He is a current recipient of an Australian National Health and Medical Research Council Leadership Investigator Grant. KS has received consultancy fees and speaker's honoraria from Baxter Healthcare and speaker's honoraria from Roche. He is on the medical advisory board of Fresenius Medical Care for Australia and New Zealand.

**Patient and public involvement** Patients and/or the public were involved in the design, or conduct, or reporting, or dissemination plans of this research. Refer to the Methods section for further details.

**Patient consent for publication** Not applicable.

**Provenance and peer review** Not commissioned; externally peer reviewed.

**Data availability statement** Not applicable.

**ORCID iDs**
Wubshet Tesfaye http://orcid.org/0000-0001-7208-2330
Vincent L. Versace http://orcid.org/0000-0002-8514-1763
Rita McMorrow http://orcid.org/0000-0002-2835-9504
Judith Fethney http://orcid.org/0000-0003-0716-596X
Natasa Gisev http://orcid.org/0000-0003-0452-8470
Ronald L. Castelino http://orcid.org/0000-0002-5128-7115

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
