## [Reviewer comments · BMJ Open]

ARTICLE DETAILS

TITLE (PROVISIONAL)	The impact of a pharmacy-led screening and intervention in people at risk of or living with chronic kidney disease in a primary care setting: A cluster randomised trial protocol
AUTHORS	Tesfaye, Wubshet; Krass, Ines; Sud, Kamal; Johnson, David; Van, C; Versace, Vincent; McMorrow, Rita; Fethney, Judith; Mullan, Judy; Tran, Anh; Robson, Breonny; Vagholkar, Sanjyot; Kairaitis, Lukas; Gisev, Natasa; Fathima, Mariam; Tong, Vivien; Coric, Natali; Castelino, Ronald Lynel

VERSION 1 – REVIEW

REVIEWER	Strumia, Mathilde CHU Limoges
REVIEW RETURNED	01-Sep-2023

GENERAL COMMENTS	Thank you for giving me the opportunity to reread the protocol. I don't think the title is clear about the type of article: could you add the word "protocol" to the title? It is possible to explain why people over 75 years are not included in the study ? Indeed, they are the people most at risk, but this age limit is not discussed.
--

REVIEWER	Almarsdottir, Anna Kobenhavns Universitet, Pharmacy
REVIEW RETURNED	13-Nov-2023

GENERAL COMMENTS	Dear Authors. Thank you for submitting this trial protocol. The trial will test a very important pharmacy-based intervention and help shed light on how community pharmacists can be further mobilized in screening activities and in improving medicines use in citizen or patient populations. The manuscript is well structured and written in clear English. The protocol is very clear in most aspects, but I have a few items that I would like adressed: 1) how come you have hypotheses about economic benefits, but do not include this as an objective? 2) you mention humanistic outcomes in your methods, but these are not in the hypotheses. 3) on page 19 you write shortly about engaging stakeholders. I would like to know more about your plans on how to involve them, at what stages, and in what way? 4) you do not delve into potential risks or limitations that can be encountered when dealing with community pharmacy, especially what is known about staff turnover and burnout in health care workers.
---

	5) there is no mention of a feasibility study. Can you address this issue, either by telling about your feasibility study, or why it has not been carried out?
--	--

VERSION 1 – AUTHOR RESPONSE

Reviewer 1

Thank you for giving me the opportunity to reread the protocol.

I don't think the title is clear about the type of article: could you add the word "protocol" to the title?

It is possible to explain why people over 75 years are not included in the study? Indeed, they are the people most at risk, but this age limit is not discussed.

Response: Thanks for your feedback. We have now modified the title to show that this is a protocol paper.

The age group we used in our study is informed by two major factors: (1) the QKidney® tool that we are using to estimate and stratify risk was developed and validated in this age range, and (2) most previous screening studies, including those assessing cost-benefit analyses on CKD screening, have largely focused on the 35-74 years age group. This focus highlights the emphasis on population groups that would most benefit from early detection and intervention approaches. We have identified some relevant references below.

1. Predicting the risk of Chronic Kidney Disease in Men and Women in England and Wales: prospective derivation and external validation of the QKidney® Scores.
2. Population-Wide Screening for Chronic Kidney Disease: A Cost-Effectiveness Analysis.
3. Predicting the risk of chronic kidney disease in the UK: an evaluation of QKidney® scores using a primary care database.
4. Cost-effectiveness of screening for chronic kidney disease in the general adult population: a systematic review.

Reviewer 2

Dear Authors. Thank you for submitting this trial protocol. The trial will test a very important pharmacy-based intervention and help shed light on how community pharmacists can be further mobilized in screening activities and in improving medicines use in citizen or patient populations. The manuscript is well structured and written in clear English. The protocol is very clear in most aspects, but I have a few items that I would like addressed:

- 1) how come you have hypotheses about economic benefits, but do not include this as an objective?
- 2) you mention humanistic outcomes in your methods, but these are not in the hypotheses.

Response: We appreciate the positive remarks. The comments raised in relation to the hypotheses and objectives have now been both addressed in the manuscript (page 4).

- 3) on page 19 you write shortly about engaging stakeholders. I would like to know more about your plans on how to involve them, at what stages, and in what way?

Response: This has now been expanded under the 'Patient and public involvement' section as follows "The study will engage key stakeholders in designing, conducting, and reporting the research, as

shown in Figure 2. Input from consumers and consumer organisations will be actively sought throughout the study implementation. This will be achieved by involving consumers and organisations, such as Kidney Health Australia and the Pharmaceutical Society of Australia, in both the project steering committee and the expert advisory group. Their roles will include contributing to the development of training materials for pharmacists, advising on effective communication and engagement strategies with pharmacists and GPs, and facilitating the implementation and dissemination of research progress and findings via newsletters.”

4) you do not delve into potential risks or limitations that can be encountered when dealing with community pharmacy, especially what is known about staff turnover and burnout in health care workers.

Response: We have now expanded on this part – please check the following in our limitations: “Finally, the high staff turnover rates and uneven distribution of the pharmacy workforce based on remoteness may affect the recruitment of pharmacists and patients in certain settings, particularly in remote and rural areas.^{50,51}”

5) there is no mention of a feasibility study. Can you address this issue, either by telling about your feasibility study, or why it has not been carried out?

Response: Thanks for this feedback. We have highlighted this towards the end of the background section, which has been reproduced here for your information: “Our pilot work in an Australian context involving 24 community pharmacies and 389 patients found that 52% of screened participants had moderate or high risk of developing CKD, but only 53% of them subsequently had kidney function tests via a pathology provider.²⁸ The pilot work highlighted issues such as poor referral uptake by GPs, lack of service remuneration, and absence of objective measurements for kidney function.^{28,29} To address these challenges, the proposed trial comprises an improved model of care that involves a more effective collaboration between pharmacists and GPs, including incentives for both, and implementing a POCT for objective measurements of kidney function.”

VERSION 2 – REVIEW

REVIEWER	Almarsdottir, Anna Kobenhavns Universitet, Pharmacy
REVIEW RETURNED	08-Dec-2023
GENERAL COMMENTS	Thank you for the revision of the manuscript. All my comments have been adressed.

VERSION 2 – AUTHOR RESPONSE